# OpenReview forum: "The Heterogeneous Safety Impacts of Benign Multilingual Fine-Tuning"
_ICML.cc/2026/Conference — ICML 2026 regular_

### Official Review · Reviewer_R5sq · 2026-02-27

**Soundness:** 3
**Presentation:** 4
**Significance:** 3
**Originality:** 3
**Overall Recommendation:** 4
**Confidence:** 2

**Summary:**

This paper presents a comprehensive empirical and mechanistic study of how benign multilingual fine-tuning affects safety alignment in small multilingual LLMs (0.6B-4B). Using LoRA fine-tuning on a 1k benign instruction dataset translated into nine languages and evaluating with both English and translated SORRY-Bench prompts, the authors show heterogeneous and sometimes drastic shifts in adversarial compliance rates that depend on both the fine-tuning language and the evaluation language. They further find that these safety changes are largely decoupled from general capabilities and provide an architecture-dependent mechanistic account via layer-wise shifts in a contrastive “safety direction.” The paper releases a Multilingual-Benign-Tune dataset and extends SORRY-Bench with multilingual prompts.

**Compliance With Llm Reviewing Policy:**

Affirmed.

**Final Justification:**

I view the overall concerns as partially resolved. I hold on to my reviews: the relatively small fine-tuning dataset.

Overall, I appreciate the authors’ clarifications and the additional results provided in the rebuttal. I am cautious about improving the overall score. To be honest, I think this work verifies an important issue, but it focuses more on superficial fine-tuning and lacks in-depth analysis.

**Key Questions For Authors:**

See weaknesses
1. How robust are the safety results to decoding choices (e.g., greedy decoding, temperature 0.2, top-p variations)? Do the cross-language amplification and architecture-specific trends persist?
2. Can you quantify the error attributable to back-translation + English-only judging by running bilingual human adjudication (or a multilingual LLM judge) on a stratified subset of local-language outputs?
3. How sensitive are the mechanistic correlations to the choice of layer, token pooling (final vs. average), and the specific adversarial/non-adversarial prompt sets? Do you observe similar trends if you derive a linear separator (e.g., logistic regression) and track its drift?
4. Beyond TinyMMLU and English perplexity, can you include a multilingual utility metric (e.g., xMMLU or task-style helpfulness in the local language) to further support the “capability–safety decoupling” claim?

**Limitations:**

yes

**Strengths And Weaknesses:**

Strengths
1. The focus on safety drift induced by benign multilingual fine-tuning is original and timely, complementing existing work on harmful fine-tuning by highlighting a realistic and under-explored failure mode.
2. The mechanistic analysis (layer-wise contrastive safety directions and displacement-compliance correlations) offers a plausible, architecture-dependent interpretation of observed behavioral drift.
3. Multiple evaluation angles (English vs. local safety, TinyMMLU, English and multilingual perplexity, non-adversarial compliance) reduce the risk of conflating safety drift with generic degradation.
4. The core claims are clearly stated and revisited throughout, with consistent visualization of cross-language effects (ratios, scatter plots).
The mechanistic section is explained at a high level with sufficient mathematical definition to follow the proposed direction-drift method.

Weaknesses
1. Safety evaluation rests on a single automated judge (SORRY-Bench autorater) and, for non-English generations, on back-translation to English via NLLB; no human adjudication is reported. This compounds two sources of potential error (generation→translation→judge).
2. The mechanistic correlations are computed on small per-model samples (eight non-English languages + English) with a post-hoc choice of the “80th-percentile” layer; multiple-testing corrections and robustness across layer choices or alternative embeddings/directions are not fully established.
3. The benign fine-tuning dataset is relatively small (1k examples) and synthetic; the generality of the results to realistic enterprise/user datasets is not demonstrated.
4. The experiments test only three series LLMs, which is not convincing enough for generalization and statistical significance.

---

> ### Author Rebuttal · Authors · 2026-03-31
>
> **We thank the reviewer for noting the timely nature of this study, and for finding our evaluation strategies rigorous and mechanistic analyses meaningful. We are grateful for the high scores on soundness, presentation, significance, and originality.**
>
> We address specific comments below:
>
> ```
> > Safety evaluation rests on a single automated judge [...]; no human adjudication is reported.
> ```
>
> It is not feasible to report human adjudication on the model outputs. We conducted over 2000 evaluations, and we did not have the resources to manually annotate these model outputs as compliance or refusal. We performed some manual verification of a subset of the outputs, presented in the table in our response to B9Wq. We will add this and other details to the paper.
>
> ```
> > The mechanistic correlations are computed on small per-model samples (eight non-English languages + English) with a post-hoc choice of the “80th-percentile” layer; multiple-testing corrections and robustness across layer choices or alternative embeddings/directions are not fully established.
> ```
>
> As the reviewer noted in their strengths section, “mechanistic analysis offers a plausible, architecture-dependent interpretation of observed behavioral drift”. Our objective with this research was to show the occurrence of changes in safety compliance rates due to benign multilingual finetuning. We leave deeper analysis of the per layer activations to future work, and showcase through the figure 4 the presence of relationships between compliance rates and internal activations. Lastly, we defer to the reviewers own description of the strengths of the mechanistic analysis to indicate our motivations for including it: “The mechanistic section is explained at a high level with sufficient mathematical definition to follow the proposed direction-drift method.”
>
> ```
> > The benign fine-tuning dataset is relatively small (1k examples) and synthetic; the generality of the results to realistic enterprise/user datasets is not demonstrated.
> ```
>
> Our paper aims to publish a benchmark dataset and evaluation pipeline. Our dataset mimics the real-world Dolly dataset . The finetuning data is limited to a realistic 1000 samples due to the cost of annotation, which itself was large enough to see safety metrics change as a result of the finetuning. We avoid the experiment results being affected by the choice of finetuning data in different languages by using non-proprietary synthetic data that is identical in all languages, verified for correctness.
>
> ```
> > The experiments test only three series LLMs, which is not convincing enough for generalization and statistical significance.
> ```
>
> We thank the reviewer for highlighting that additional experiments would make the claims stronger. Our results so far show that generalization cannot be expected, using over 2000 experiment runs. We have now added additional experiments with larger models. Llama, Gemma, and Qwen are popular series of text-only open source models, and would welcome any specific model suggestions from the reviewer that they would like to see included in our final paper.
>
> ```
> > How robust are the safety results to decoding choices (e.g., greedy decoding, temperature 0.2, top-p variations)? [...]
> ```
>
> We test with several finetuning hyperparameter values. For model inference, we adopt the sorry_bench pipeline exactly. Our objective here is to replicate the sorry_bench autorater as closely as possible, so we do not make any modifications (unless necessary, like the NLLB-200 translation step).
>
> ```
> >  Can you quantify the error attributable to back-translation + English-only judging by running bilingual human adjudication (or a multilingual LLM judge) on a stratified subset of local-language outputs?
> ```
>
> We verified a subset of the model outputs for translation accuracy, and found that the NLLB-200 translation was mostly accurate. Detailed stats are presented in our response to reviewer B9Wq.
>
> ```
> > Beyond TinyMMLU and English perplexity, can you include a multilingual utility metric (e.g., xMMLU or task-style helpfulness in the local language) to further support the “capability–safety decoupling” claim?
> ```
>
> As argued by the TinyBenchmarks authors, TinyMMLU is designed to provide equivalent scores to (x)MMLU with fewer datapoints. Line 308 includes details about our multilingual evaluations, which we will clarify further. We also include Local Language Perplexity evaluation results in Figure 9 (Appendix), with only Irish showing consistently significantly varied perplexity after fine-tuning.
>
> **We thank the reviewer for their careful feedback and comments, and look forward to their response to our clarifications above.**

---

> > ### Author Rebuttal · Reviewer_R5sq · 2026-04-03
> >
> > Thank you to the authors for the detailed rebuttal and for engaging carefully with the questions in my review.
> >
> > The rebuttal addresses several of my main concerns. In particular, the clarification regarding human adjudication of the data, the decoding choices of hyperparams, may help resolve important gaps in the original submission.
> >
> > I view the overall concerns as partially resolved or unresolved. I hold on my reviews below:
> > (a) the relatively small fine-tuning dataset; (b) limited diversity of tested LLMs; (c) the error brought by back-translation.
> >
> > Overall, I appreciate the authors’ clarifications and the additional results provided in the rebuttal. I therefore consider the concerns partially resolved and keep my overall assessment unchanged.

---

> > > ### Author Response · Authors · 2026-04-08
> > >
> > > We thank the reviewer for their feedback. We have replied directly to the concerns highlighted in the acknowledgement in our previous response, which we repeat below:
> > >
> > > a) **dataset size**:
> > > Our dataset size is limited by the ability to annotate samples across 8 languages, and is sufficient to show the impacts of multilingual finetuning on model refusal rates. Further, our dataset is much larger than standard recommendations from OpenAI (50-100 samples recommended, we use 1000) (https://developers.openai.com/api/docs/guides/supervised-fine-tuning), which we believe makes it a meaningful contribution to the research community.
> > >
> > > b) **LLM diversity**:
> > > We have included additional experiments to improve diversity of LLMs used in the experiment, and would like to gently repeat our request to the reviewer to specify which LLMs they would have expected to see in this evaluation.
> > >
> > > c) **potential back-translation errors**:
> > > We measured LLM back translation errors and referred to this in our response. Please see the table in our response to reviewer B9Wq for detailed statistics. It is not feasible to verify translations manually for every run (2000+ runs, 880 samples evaluated per language per run), so we use a sample here.
> > >
> > >
> > > Considering the significant additional experiments added to the paper during the rebuttal period, we would appreciate more clarity from the reviewer about their acknowledgement indicating “concern(s about) the core tenets of the work,” addressing which ”requires … significant update(s) to the paper.”

---

### Official Review · Reviewer_CAdp · 2026-03-09

**Soundness:** 2
**Presentation:** 3
**Significance:** 2
**Originality:** 3
**Overall Recommendation:** 4
**Confidence:** 3

**Summary:**

In this manuscript, the authors mention that prior studies have demonstrated that fine-tuning on benign datasets may still undermine the safety alignment of LLMs. However, most of these studies have primarily focused on English. Therefore, the authors conduct a comprehensive evaluation of the impact of fine-tuning with benign data on model safety in a multilingual setting. Specifically, experiments are performed on models ranging from 0.6B to 4B parameters across eight languages. The topic is of considerable significance for advancing large language model training and safety alignment. However, several aspects require further clarification and improvement to enhance the overall rigor and completeness of the paper.

**Compliance With Llm Reviewing Policy:**

Affirmed.

**Final Justification:**

My concerns have been addressed, and I will raise my score to 4.

**Key Questions For Authors:**

- The conclusions in the paper are drawn from experiments on small models ranging from 0.6B to 4B parameters, which is insufficient. Larger models, typically above 7B parameters, are more widely used and generally exhibit stronger multilingual alignment capabilities. It is therefore unclear whether these conclusions hold for more general, larger models.
- Most experiments are conducted with only 1 training epoch. Although the appendix mentions that results with 3 epochs do not differ significantly, in Figure Qwen3-0.6B there is still a noticeable difference between 1 and 3 epochs. It is recommended that the authors test with more epochs, such as 5 or 8, because using only 1 and 3 epochs does not sufficiently demonstrate that the results are insensitive to epoch count.
- In Figure 5, we can observe that small models (e.g., 1B) show considerable safety impact, whereas in larger models (e.g., Gemma-3-4B), the Adversarial Compliance after fine-tuning across different languages is almost identical to the base model, and in some cases even slightly lower. This suggests that the conclusions drawn from small models may not generalize to larger models. Experiments on larger models are recommended.
- Minor issues:
  1. Many figures have fonts that are too small, e.g., Figures 2 and 4.
  2. The evaluation metrics used should be explained in the experimental setup section.

**Limitations:**

Yes

**Strengths And Weaknesses:**

**Strengths:**
- Evaluating the impact of benign data fine-tuning on model safety across multiple languages is a valuable starting point, which can provide useful insights and support for subsequent research.
- The authors evaluate more than 350 model variants, covering multiple languages and different model architectures.

 **Weaknesses:**
- The experiments are conducted only on small models ranging from 0.6B to 4B parameters, which is insufficient, as larger models generally exhibit stronger multilingual alignment capabilities.
- The SORRY-Bench-Multilingual benchmark is not publicly released.

---

> ### Author Rebuttal · Authors · 2026-03-31
>
> **We thank the reviewer for their feedback and for recognising the merit of the problem setting being studied. We would like to clarify that the multilingual fine-tuning and evaluation data (reviewed and corrected by expert human annotators) was submitted as part of our supplementary materials, and will be released with an open-source license as part of our paper. We thank the reviewer for pointing out that we had not made this clear in the writing.**
>
> Specific concerns are addressed below:
>
> ```
> > The SORRY-Bench-Multilingual benchmark is not publicly released.
> ```
> The finetuning and evaluation data is included in our supplementary materials, and will be released publicly with our paper. We thank the reviewer for bringing this to our attention and will clarify this further in the writing.
>
> ```
> > The conclusions in the paper are drawn from experiments on small models ranging from 0.6B to 4B parameters, which is insufficient. [...]
> ```
>
> We thank the reviewer for pointing out the importance of experimenting with larger models. While initially limited by compute availability, we have now expanded our experiments to include the following models (1 seed and epoch each). We plan to include all results (in accordance with the hyperparameter ablations discussed in the final paper):
> Llama-3-8B
> Qwen3-8B
> Qwen3-14B
> Qwen3-32B
> Results for these models can be seen in our response to review Qw4u. We thank the reviewer for explicitly mentioning this feedback and encouraging us to procure compute to conduct these additional experiments.
>
> ```
> > Most experiments are conducted with only 1 training epoch. [...] It is recommended that the authors test with more epochs [...]
> ```
>
> We acknowledge the reviewer prudently highlighting that our epochs hyperparameter sweep could be larger. Again, we have so far been constrained by compute, but will be able to include these results in the final paper. We will use the suggested sweep and vary epochs from 1 through 8 instead of 1 through 3.
>
> ```
> > In Figure 5, we can observe that small models (e.g., 1B) show considerable safety impact [...]. This suggests that the conclusions drawn from small models may not generalize to larger models. [...]
> ```
>
> We continue to observe the heterogeneous impact of the finetuning on model safety with larger models. With the larger Llama model, for instance, we find that compliance rates increase, whereas with the larger Qwen model, they decrease. The claim we wish to make with this paper is that compliance rates in English and in the local language can change when the model is finetuned in that language, and this seems to be true for the larger models too.
>
> ```
> > Many figures have fonts that are too small, e.g., Figures 2 and 4.
> > The evaluation metrics used should be explained in the experimental setup section.
> ```
>
> We will improve our figures with bigger fonts and better formatting. We will also better document our evaluation metric in the experimental section: we have adapted the sorry-bench autorater to measure compliance rates. When outputs are in English we use the autorater as is, and when they are in other languages we first translate the output to English using NLLB-200, and then adopt the same autorater. The accuracy of this pipeline was verified by evaluating a subset of the model outputs from one run. (It was prohibitively expensive to get annotators for 880 model outputs for 2000+ runs). We believe our refusal detection pipeline to be on par with sorry_bench, and will release the sorry_bench_multilingual benchmark for extensive community verification and future work on this.
>
> **We are grateful for the constructive and actionable feedback provided by the reviewer. Noting the explicit mention to use the review scores sparingly, and our clarifications about 1) releasing our human-annotated multilingual safety benchmark sorry-bench-multilingual and 2) experiments with models up to 32B in size, we would be grateful if the reviewer would please consider revising their review score.**

---

> > ### Author Rebuttal · Reviewer_CAdp · 2026-04-04
> >
> > Thank you for the response.
> >
> > First, the authors have conducted experiments on 8B, 14B, and 32B models, which addresses one of my concerns. However, I still have the following issues. For example, the main experiments rely only on the minimum number of training epochs. In addition, some concerns raised by other reviewers remain unaddressed, such as the relatively small size of the fine-tuning dataset. Therefore, I will maintain my current score.

---

> > > ### Author Response · Authors · 2026-04-08
> > >
> > > We thank the reviewer for their review and partial acknowledgement. We include below additional results for epochs 5 and 8, as suggested by the reviewer. These continue to demonstrate the heterogenous impacts of multilingual finetuning. The average change in english compliance rates ratio from 5 to 8 epochs is 8.2%, and average change in translated compliance rate ratio from 5 to 8 epochs is just 1.6%, showing that additional epochs do not meaningfully change the results obtained.
> > >
> > > Regarding the dataset size, we are constrained by the data we were able to annotate through human review across 8 languages. We believe that this dataset serves as a meaningful contribution to the community. We would also like to point out that while seemingly minimal, finetuning using this “small” dataset is enough to meaningfully change refusal rates. Further, our dataset has 1000 rows, which far exceeds industry standards (OpenAI recommends 50-100 datapoints as a meaningful finetuning dataset size for SFT: https://developers.openai.com/api/docs/guides/supervised-fine-tuning).
> > >
> > > |model and finetune language|[5 epochs] English Compliance Rate Ratio: finetuned/initial|[5 epochs] Local Language Compliance Rate Ratio: finetuned/initial|[8 epochs] English Compliance Rate Ratio: finetuned/initial|[8 epochs] Local Language Compliance Rate Ratio: finetuned/initial|
> > > |-|-|-|-|-|
> > > |Gemma3-1B-DA |2.23 |2.81 |2.36 |2.86 |
> > > |Gemma3-1B-EL |1.56 |2.28 |1.58 |2.46 |
> > > |Gemma3-1B-ES |2.31 |3.32 |2.36 |4.19 |
> > > |Gemma3-1B-GA |1.69 |0.38 |2.03 |0.62 |
> > > |Gemma3-1B-HI |2.15 |2.76 |2.38 |2.66 |
> > > |Gemma3-1B-PT |2.33 |2.54 |2.36 |2.9 |
> > > |Gemma3-1B-TL |2.22 |2.26 |2.22 |2.24 |
> > > |Gemma3-1B-ZH |2.18 |3.93 |2.41 |3.37 |
> > > |Gemma3-4B-DA |0.93 |2.47 |1.12 |3.79 |
> > > |Gemma3-4B-EL |0.77 |1.85 |0.73 |2.42 |
> > > |Gemma3-4B-ES |0.99 |3.46 |1.28 |4.28 |
> > > |Gemma3-4B-GA |0.86 |0.87 |0.9 |1.02 |
> > > |Gemma3-4B-HI |1.04 |2.77 |1.17 |3.21 |
> > > |Gemma3-4B-PT |1.04 |2.61 |1.11 |3.1 |
> > > |Gemma3-4B-TL |0.95 |2.19 |0.97 |2.48 |
> > > |Gemma3-4B-ZH |0.94 |2.64 |1.13 |3.63 |
> > > |Llama3.2-1B-DA |1.34 |1.15 |1.29 |1.14 |
> > > |Llama3.2-1B-EL |1.27 |1.01 |1.26 |0.7 |
> > > |Llama3.2-1B-ES |1.33 |1.87 |1.33 |1.82 |
> > > |Llama3.2-1B-GA |1.1 |0.61 |1.23 |1 |
> > > |Llama3.2-1B-HI |1.33 |0.75 |1.5 |0.48 |
> > > |Llama3.2-1B-PT |1.38 |1.83 |1.62 |1.93 |
> > > |Llama3.2-1B-TL |1.14 |1.01 |1.1 |1.22 |
> > > |Llama3.2-1B-ZH |1.29 |1.99 |1.35 |1.68 |
> > > |Llama3.2-3B-DA |1.26 |0.91 |1.13 |0.65 |
> > > |Llama3.2-3B-EL |1.13 |0.37 |1.3 |0.35 |
> > > |Llama3.2-3B-ES |1.2 |2.48 |1.29 |2.89 |
> > > |Llama3.2-3B-GA |1.19 |0.67 |1.09 |0.82 |
> > > |Llama3.2-3B-HI |1.33 |1.42 |1.09 |0.53 |
> > > |Llama3.2-3B-PT |1.36 |1.9 |1.41 |2.75 |
> > > |Llama3.2-3B-TL |1.13 |0.73 |1.09 |0.7 |
> > > |Llama3.2-3B-ZH |1.06 |1.43 |1.09 |1.11 |
> > > |Qwen3-0.6B-DA |0.72 |0.32 |0.76 |0.42 |
> > > |Qwen3-0.6B-EL |0.91 |0.35 |0.91 |0.29 |
> > > |Qwen3-0.6B-ES |0.78 |0.73 |0.86 |0.73 |
> > > |Qwen3-0.6B-GA |0.6 |1.03 |0.62 |0.79 |
> > > |Qwen3-0.6B-HI |0.68 |0.19 |0.74 |0.14 |
> > > |Qwen3-0.6B-PT |0.84 |0.73 |0.84 |0.74 |
> > > |Qwen3-0.6B-TL |0.78 |0.32 |0.72 |0.28 |
> > > |Qwen3-0.6B-ZH |0.87 |1.09 |0.89 |1.02 |
> > > |Qwen3-4B-DA |0.8 |0.74 |0.72 |0.92 |
> > > |Qwen3-4B-EL |0.18 |0.54 |0.54 |0.08 |
> > > |Qwen3-4B-ES |0.92 |1.34 |0.99 |1.32 |
> > > |Qwen3-4B-GA |0.46 |0.27 |0.49 |0.33 |
> > > |Qwen3-4B-HI |0.8 |0.79 |0.43 |0.12 |
> > > |Qwen3-4B-PT |0.94 |1.53 |1.02 |1.28 |
> > > |Qwen3-4B-TL |0.32 |0.72 |0.41 |0.64 |
> > > |Qwen3-4B-ZH |1.01 |2.13 |1.04 |2.41 |

---

### Official Review · Reviewer_B9Wq · 2026-03-13

**Soundness:** 2
**Presentation:** 2
**Significance:** 3
**Originality:** 3
**Overall Recommendation:** 3
**Confidence:** 4

**Summary:**

This paper investigates how fine-tuning LLMs with benign (non-adversarial) data in different languages affects safety alignment. The authors LoRA fine-tune six model variants on a 1,000-example instruction dataset translated into nine languages, then evaluate adversarial compliance in both English and the fine-tuning language. The paper provides several analyses and findings regarding the safety draft phenomenon in the multilingual setting. The authors also provide a multilingual fine-tuning dataset and a translated SORRY-Bench suite.

**Compliance With Llm Reviewing Policy:**

Affirmed.

**Key Questions For Authors:**

Based on the experiments and analyses, what are the potential mitigation strategies for this issue?

Is it rigorous enough to use PCA to project model parameters into a 2D space? Will it cause any potential information loss during the compression?

**Limitations:**

Yes

**Strengths And Weaknesses:**

Strengths

- The paper discusses a very interesting and practically important problem. The findings, such as benign multilingual fine-tuning has heterogeneous safety impacts, are meaningful and critical.

- The paper provides several in-depth analysis on the safety draft issues and provides several valuable insights.

- The paper also provides a multilingual dataset and evaluation suite, which would benefit the research community.

Weaknesses

- The paper only conducted limited experiments on small-scale models and LoRA fine-tuning. From an academic research perspective, it’s critical to conduct the empirical studies on full-finetuning settings and on larger-scale models to validate the generalizability of the findings.

- It could introduce potential biases or error sources when purely relying on automatic translation (via Gemini 2.5 Flash). It would be more rigorous if there were experiments on parallel datasets written by a native human speaker, or at least, the auto translation should be rigorously validated by humans.

- The paper would benefit from more in-depth analysis to identify or help explain the root cause of the phenomenon. For example, it would be critical to investigate which safety categories are most vulnerable, whether certain language features, e.g., morphological complexity, or script type, could predict or correlate with the drift patterns.

- The presentation could be further improved. For example, how to interpret the mechanistic analysis in Section 6, and what implications could be drawn, are not clearly presented. Several analysis figures are difficult to interpret at a glance.

---

> ### Author Rebuttal · Authors · 2026-03-31
>
> **We thank the reviewer for noting our papers goals and for highlighting the importance and novelty of the problem studied. We also appreciate the recognition of the benchmark and multilingual datasets produced as part of this research, which will all be open-sourced and made available for reuse by the research community for further study.**
>
> We address specific concerns raised by the reviewer below:
>
> ```
> > The paper only conducted limited experiments on small-scale models and LoRA fine-tuning.
> ```
>
> We acknowledge the limitations raised by the reviewer about results being limited to small models. While this was initially due to computational constraints, we have since expanded our experiments to include the larger models recommended (results included in the table in our response to reviewer Qw4u):
> * Llama-3-8B
> * Qwen3-8B
> * Qwen3-14B
> * Qwen3-32B
>
> While these results currently only include finetuning runs with a single seed and epoch, we will complete these experiments and include the findings in the final paper. Here we provide new results to include in Figure 2, reporting the ratios of compliance rates in English and the local language, between finetuned and base models
>
> ```
> > It could introduce potential biases or error sources when purely relying on automatic translation (via Gemini 2.5 Flash). It would be more rigorous if there were experiments on parallel datasets written by a native human speaker, or at least, the auto translation should be rigorously validated by humans.
> ```
>
> All evaluation datasets produced as part of this research went through a rigorous annotation process. Human reviewers reviewed and corrected all input prompts used to test model safety in various languages, and the compliance rates we report are on these corrected prompt inputs. These are included in the supplementary material with our submission, and will be open-sourced to the community along with our paper. We believe this is a highly valuable contribution for future research and would like to request the reviewer to please update their score given this clarification.
>
> The NLLB translation is applied to the outputs, as part of our evaluation pipeline, since the Sorry-bench autorater was only validated in English. While it was not feasible to review all the outputs produced by our experiments (over 2000 runs, each with 880 model outputs in each non-english language) we present some results from a manual verification that prompted us to trust that the accuracy of this pipeline is similar to sorry bench:
>
> | (manual verification on a 50 sample subset) | Base Model | Hindi FineTuned Model |
> |---|---|---|
> | AutoRater Judgement = Refuse | 46/50 | 50/50 |
> | AutoRater Judgement = Comply | 47/50 | 49/50 |
>
> ```
> > The presentation could be further improved. [...]
> ```
>
> We thank the reviewer for sharing this feedback, and we will improve the figures, captions, and discussion of results, as noted in this and other reviewer responses.
>
> ```
> > Based on the experiments and analyses, what are the potential mitigation strategies for this issue?
> ```
>
> Mitigation can only occur if model deployers are aware of this issue, and can measure it. No existing papers have addressed this issue or proposed to measure it. Our fully-automated benchmark provides a tool for model deployers to measure multilingual safety, and prevent deployment when the behavior is unacceptable.
>
> ```
> > Is it rigorous enough to use PCA to project model parameters into a 2D space? Will it cause any potential information loss during the compression?
> ```
>
> In figure 4, we use PCA purely as a visualisation tool. Our method and evaluation does not use PCA, and therefore the metrics we report are immune to information loss from PCA.
>
> **We thank the reviewer for their feedback and encouragement for us to expand our results to additional (larger) models. Further, we would also like to clarify that our multilingual evaluation datasets are all reviewed and rewritten (where required) by human annotators, and model outputs are auto-translated  through NLLB-200, which we found to be accurate based on a subset of data studied (the table here shows negligible impact of translation quality on the Cohen-Kappa measured by the sorry-bench autorater) We will improve the writing to clarify this information, and would be grateful if the reviewer please considered updating their scores accordingly.**

---

> > ### Author Rebuttal · Reviewer_B9Wq · 2026-04-03
> >
> > N/A

---

> > > ### Author Response · Authors · 2026-04-08
> > >
> > > We thank the reviewer for their acknowledgement and detailed and constructive review. Considering that all concerns have been mitigated, we would be grateful if the score could be revised accordingly. Currently, the response acknowledges that all concerns have been addressed, but the score has remained unchanged.

---

### Official Review · Reviewer_Qw4u · 2026-03-19

**Soundness:** 4
**Presentation:** 3
**Significance:** 3
**Originality:** 3
**Overall Recommendation:** 5
**Confidence:** 3

**Summary:**

The paper explores how finetuning a model in a given language (they study English, Chinese, Danish, Greek, Hindi, Irish, Portuguese, Spanish and Tagalog) on a benign dataset can negatively affect the model's adversarial robustness in English. The authors present a plethora of results across various axes including model family, model size, capability and safety, mechanistic explanations, and beyond.

**Compliance With Llm Reviewing Policy:**

Affirmed.

**Final Justification:**

I thought it was a solid paper, and I remain of that view. Nothing changed too much in the review period, happy to hear that the authors will incorporate my suggestions.

**Key Questions For Authors:**

* on page 3, you say that translations were reviewed and rewritten by native speakers. Were all of them, or just a subset? I'm additionally confused because on page 7 you say that only a subset were checked. If you only checked a subset, how do you know your data are ok quality?

* since you used google translate to get the english into the other languages, why not use google translate to go back from those languages into english (instead using NLLB-200)? seems weird to use it in one direction but not the other?

* does TinyAlpacaEval have a citation?

* can you increase the size of the dots and lines in Figure 1? It's quite hard to see the starting empty dots in a printed copy of the paper. Also, does 4b get less compliant over time? Why do you think that is?

* text in Figure 2 is too small to read in my printed version, can you please increase it

* what is Figure 4 trying to tell us? I found it quite hard to understand, maybe could be made more clear in the caption and esp in the text? Also the text is once again too small to read on my printed copy.

**Limitations:**

yes

**Strengths And Weaknesses:**

## Soundness

The paper appears technically sound. Extensive experiments were performed, results were discussed clearly.

## Presentation

Overall the paper is very well-presented. The plots could be improved (see questions). But very solid overall here, a pleasure to read.

## Significance

While there is no single obviously high-impact result in the paper, that does not hold it back from being a useful reference, since it is a somewhat extensive initial study (admittedly on small models) and to my knowledge the first explicit demonstration of the variability of impact of multilingual finetuning.

## Originality

None of the concepts are new, but explicitly studying this topic the way the authors do seems new to me. I think it's perfectly fine.

---

> ### Author Rebuttal · Authors · 2026-03-31
>
> **We thank the reviewer for their thorough review and feedback. This research aims to both showcase the impact of benign multilingual finetuning on safety (measured in English and in other languages), as well as to provide a replicable benchmark and evaluation strategy for researchers and practitioners to measure the safety of models in various languages before deployment. Our results motivate the importance of safety testing in diverse languages, and we provide a multilingual pipeline to enable this based on the established sorry-bench benchmark.**
>
> We address specific questions below:
>
> ```
> > on page 3, you say that translations were reviewed and rewritten by native speakers. Were all of them, or just a subset? …
> ```
>
> We will further clarify the writing to disambiguate this. On page 3, we state that all 440 evaluation prompt translations were verified for correctness. This is correct. On page 7, we note that a subset of the finetuning data were checked for translation quality. The finetuning data is different from the evaluation data, which we will clarify further in the final draft of the paper.
>
> ```
> > since you used google translate to get the english into the other languages, why not use google translate to go back from those languages into english (instead using NLLB-200)? …
> ```
>
> LLMs (like NLLB-200) often refused to translate the (harmful) input prompts from sorry-bench, which led us to use the API to translate these inputs. All of these translations were manually verified and corrected where needed.
>
> We used NLLB-200 to translate model outputs because it was fairly accurate (table included in response to B9Wq), and allowed us to publish a benchmark evaluation protocol that could be reused easily by the community.
>
> ```
> > does TinyAlpacaEval have a citation?
> > can you increase the size of the dots and lines in Figure 1? [...]
> > text in Figure 2 is too small to read in my printed version [...]
> ```
>
> We will add all missing citations, as well as improve the readability of the figures by increasing font sizes, line widths, and marker sizes.
>
> ```
> >  Also, does 4b get less compliant over time? Why do you think that is?
> ```
>
> Yes, training for additional epoch makes Qwen-4B less compliant in English. Appendix A.2 shows the impact of increasing training epochs for EN-language evals, and A.3 for local language evals. We see that increasing training epochs from 1 to 3 for Qwen-4B generally decreases compliance in EN evals with no trend seen for local evals.
>
> ```
> > what is Figure 4 trying to tell us? I found it quite hard to understand, maybe could be made more clear in the caption and esp in the text?  Also the text is once again too small to read on my printed copy.
> ```
>
> Figure 4 aims to illustrate that model drift in activations cannot be used naively as a proxy for inferring changes in compliance rates. This preliminary analysis relates the changes in model compliance with the internal representations learned by the model. We show that the relationship differs between models indicating different mechanisms for the shift. We will improve the readability and font sizes in this and other figures.
>
> **We are grateful to the reviewer for their comments, and would be happy to incorporate any further feedback to improve the readability and quality of our paper.**
>
> --------------
>
> We would also like to share that we have now procured additional compute and augmented our results with additional experiments. Extending figure 2 (results from finetuning over 1 epoch and seed):
>
>
>
> | model_finetune | English Compliance Rate Ratio: finetuned/initial | Local Language Compliance Rate Ratio: finetuned/initial |
> |---|---|---|
> | llama-3.1-8b-DA | 1.25 | 1.24 |
> | llama-3.1-8b-EL | 0.92 | 0.96 |
> | llama-3.1-8b-ES | 1.16 | 2.36 |
> | llama-3.1-8b-GA | 1.1 | 1.04 |
> | llama-3.1-8b-HI | 1.17 | 1.11 |
> | llama-3.1-8b-PT | 1.15 | 1.52 |
> | llama-3.1-8b-TL | 0.95 | 0.78 |
> | llama-3.1-8b-ZH | 0.86 | 1.27 |
> | Qwen3-8B-DA | 0.74 | 0.72 |
> | Qwen3-8B-EL | 0.87 | 1.04 |
> | Qwen3-8B-ES | 0.75 | 1.04 |
> | Qwen3-8B-GA | 0.87 | 0.35 |
> | Qwen3-8B-HI | 0.7 | 0.68 |
> | Qwen3-8B-PT | 0.94 | 0.83 |
> | Qwen3-8B-TL | 0.83 | 0.87 |
> | Qwen3-8B-ZH | 0.47 | 1.31 |
> | Qwen3-14B-DA | 0.69 | 0.73 |
> | Qwen3-14B-EL | 0.79 | 1.11 |
> | Qwen3-14B-ES | 0.73 | 1.29 |
> | Qwen3-14B-GA | 0.48 | 0.47 |
> | Qwen3-14B-HI | 0.85 | 0.87 |
> | Qwen3-14B-PT | 0.7 | 1.06 |
> | Qwen3-14B-TL | 0.8 | 1.17 |
> | Qwen3-14B-ZH | 0.63 | 1 |
> | Qwen3-32B-DA | 0.98 | 0.44 |
> | Qwen3-32B-EL | 1.21 | 0.85 |
> | Qwen3-32B-ES | 0.91 | 0.55 |
> | Qwen3-32B-GA | 0.93 | 0.4 |
> | Qwen3-32B-HI | 1.1 | 0.71 |
> | Qwen3-32B-PT | 0.77 | 0.62 |
> | Qwen3-32B-TL | 1.08 | 0.6 |
> | Qwen3-32B-ZH | 0.44 | 0.84 |

---

> > ### Author Rebuttal · Reviewer_Qw4u · 2026-04-04
> >
> > Thank you for answering my questions and good luck with the submission.

---

> > > ### Author Response · Authors · 2026-04-08
> > >
> > > We thank the reviewer for their effort and careful review. We will incorporate all their feedback to improve the writing and figures in the final draft.

---

### Decision · Program_Chairs · 2026-04-30

**Decision:**

Accept (regular)

**Comment:**

The reviews were borderline, but many concerns raised (eg full fine-tuning) have been resolved and remaining concerns like "evaluate on more LLMs" seem vague and unnecessary. The paper brings up an interesting analysis on multilingual safety after finetuning. The authors should add the full finetuning results on larger models (multiple seeds/epochs) as promised in the camera ready,